# Mechanistic insights into the regulation of plant phosphate homeostasis by the rice SPX2 – PHR2 complex

Zeyuan Guan [1,3], Qunxia Zhang[1,3], Zhifei Zhang[1,3], Jiaqi Zuo[1], Juan Chen[1], Ruiwen Liu[1], Julie Savarin [2], Larissa Broger[2], Peng Cheng[1], Qiang Wang[1], Kai Pei[1], Delin Zhang[1], Tingting Zou [1], Junjie Yan[1], Ping Yin [1], Michael Hothorn [2] & Zhu Liu [1✉]

Phosphate (Pi) starvation response (PHR) transcription factors play key roles in plant Pi homeostasis maintenance. They are negatively regulated by stand-alone SPX proteins, cellular receptors for inositol pyrophosphate (PP-InsP) nutrient messengers. How PP-InsP-bound SPX interacts with PHRs is poorly understood. Here, we report crystal structures of the rice SPX2/InsP$_6$/PHR2 complex and of the PHR2 DNA binding (MYB) domain in complex with target DNA at resolutions of 3.1 Å and 2.7 Å, respectively. In the SPX2/InsP$_6$/PHR2 complex, the signalling-active SPX2 assembles into a domain-swapped dimer conformation and binds two copies of PHR2, targeting both its coiled-coil (CC) oligomerisation domain and MYB domain. Our results reveal that the SPX2 senses PP-InsPs to inactivate PHR2 by establishing severe steric clashes with the PHR2 MYB domain, preventing DNA binding, and by disrupting oligomerisation of the PHR2 CC domain, attenuating promoter binding. Our findings rationalize how PP-InsPs activate SPX receptor proteins to target PHR family transcription factors.

[1] National Key Laboratory of Crop Genetic Improvement, Hubei Hongshan Laboratory, Huazhong Agricultural University, Wuhan 430070, China. [2] Structural Plant Biology Laboratory, Department of Botany and Plant Biology, University of Geneva, Geneva 1211, Switzerland. [3] These authors contributed equally: Zeyuan Guan, Qunxia Zhang, Zhifei Zhang. ✉email: liuzhu@mail.hzau.edu.cn

**P**hosphorus is an essential nutrient limiting plant growth, development, and crop productivity[1,2]. Plants have developed sophisticated signal systems to perceive, uptake, transport, and store phosphate (Pi) for the maintenance of Pi homeostasis[3–10]. During Pi limitation, a broad range of Pi starvation induced (PSI) genes are expressed in response to the nutrient deficiency[11–14]. Central regulators responsible for the transcriptional activation of PSI genes are the highly conserved Pi starvation response (PHR) family transcription factors[14–19].

Under Pi starvation, rice (*Oryza sativa*) PHR2 (OsPHR2) binds to a *cis*-element (P1BS) in the promoter of various PSI genes and up-regulates their transcription, thus optimizing rice Pi acquisition and utilisation[20–22]. In contrast, under Pi sufficient conditions, an inositol pyrophosphates (PP-InsPs) dependent negative regulator, the stand-alone SPX (SYG1/Pho81/XPR1) protein, binds to OsPHR2 and inactivates its transcriptional activity[20,21,23,24] (Supplementary Fig. 1). Despite extensive studies on the rice PHR signalling pathway, how the SPX receptor proteins sense the Pi-level correlated PP-InsPs signal and transduce this signal into transcription repression remains largely unknown.

To reveal this underlying mechanism at molecular level, here we report the crystal structure of the rice SPX2/InsP$_6$/PHR2 complex at 3.1 Å resolution, using InsP$_6$ (phytic acid) as a commercially available surrogate for the bioactive PP-InsP. In the SPX2/InsP$_6$/PHR2 complex, the InsP$_6$ is perceived by a domain-swapped SPX2 dimer. The signalling-active SPX2 dimer binds two copies of PHR2, and each polypeptide of the SPX2 dimer binds to the PHR2 MYB domain and PHR2 CC domain, respectively. Furthermore, we determine the crystal structure of the PHR2 MYB domain in complex with its target DNA at 2.7 Å resolution, to elucidate how the InsP$_6$-bound SPX2 association of PHR2 inhibits its DNA binding activity. Structural comparison, biochemical and biophysical analysis show that SPX2 senses the InsP$_6$ / PP-InsP nutrient messenger to repress the PHR2 transcriptional activity by establishing severe steric clashes with the PHR2 MYB domain, preventing DNA binding, and by disrupting the CC domain oligomerisation motif in PHR2, attenuating its transcriptional activity.

## Results

**Structure of the rice SPX2/InsP$_6$/PHR2 complex.** To determine the crystal structure of rice SPX2 in complex with rice PHR2 and InsP$_6$, we first constructed a PHR2 boundary PHR2$^{225–362}$ containing the MYB domain and the CC domain[16,25,26]. By co-expressing His-tagged PHR2$^{225–362}$ with SPX2 following Ni-NTA pull-down assay, we found that the interaction between PHR2$^{225–362}$ and SPX2 is InsP$_6$ dependent (Supplementary Fig. 2), in agreement with previous reports[23,27]. Initial crystals of the of the SPX2/InsP$_6$/PHR2$^{225–362}$ ternary complex diffracted only to low resolutions. We next identified poorly conserved loop regions in rice SPX2 (residues 36-69 and 191-280; Supplementary Fig. 3). Deletion of non-conserved insertions represents one promising approach to improve protein stability and crystallizability[28]. Based on this we expressed an engineered SPX2$^{1-202/Δ47-59}$ construct fused to the C-terminus of the macro domain of human histone mH2A1.1$^{181–366}$ for carrier-driven crystallization[28]. The crystal structure of the mH2A1.1$^{181–366}$-tagged SPX2$^{1-202/Δ47-59}$/InsP$_6$/PHR2$^{225–362}$ ternary complex was subsequently determined at 3.1 Å resolution, using InsP$_6$ (phytic acid) as a commercially available surrogate for the bioactive PP-InsPs[23,26] (Fig. 1a, Supplementary Fig. 4, and Supplementary Table 1).

In the complex structure, two InsP$_6$ molecules bind to a SPX2 dimer, and two copies of PHR2 wrap around this dimer. In order to check and assess if the mH2A1.1$^{181–366}$ fusion tag and the loop truncations affected the SPX2 structure and its ability to bind

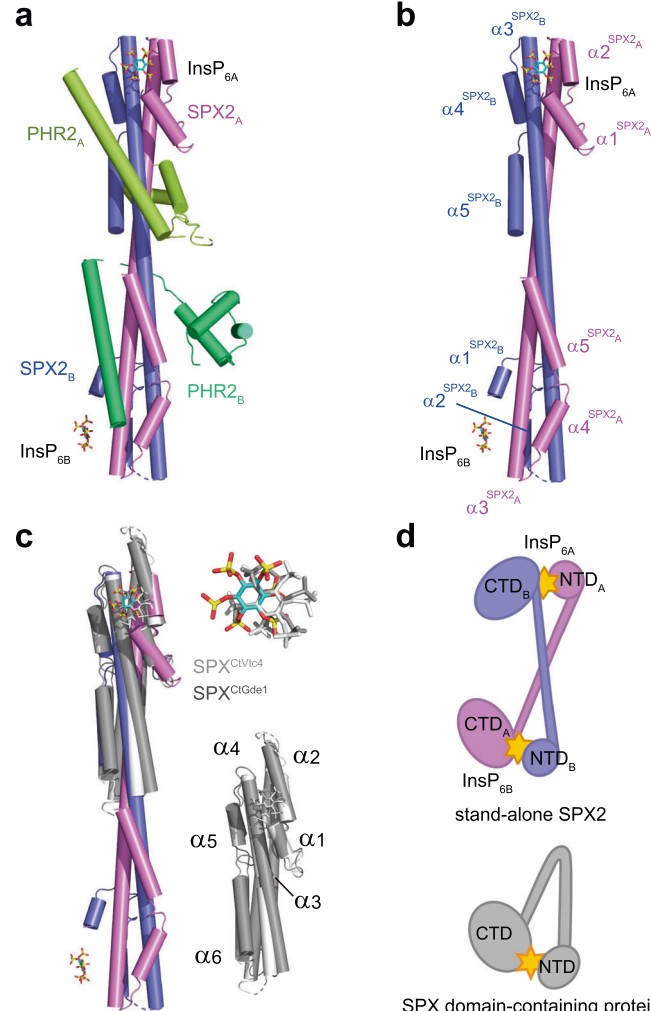

**Fig. 1 The structure of rice SPX2/InsP$_6$/PHR2 complex and topology differences between the rice stand-alone SPX2 and SPX domain-containing proteins. a** Overall structure of the ternary complex. The protomers of the two SPX2 and the two PHR2 molecules are colored in magenta, blue, yellow, and green cartoon representation, respectively. InsP$_6$ molecules are represented in sticks. **b** Domain-swapped conformation of the SPX2 dimer in the SPX2/InsP$_6$/PHR2 complex. The α-helices of each protomer are numbered and highlighted in magenta and blue, respectively. PHR2 has been omitted for clearity. **c** Structural comparison between the stand-alone SPX2 dimer and other monomeric SPX domain-containing proteins. SPX domain-containing proteins are colored in gray with numbered secondary structures. PHR2 has been omitted for clearity. **d** Topological diagrams of the stand-alone SPX2 other SPX domain-containing protein. InsP$_6$ is represented in yellow star.

PHR2, we performed analytical ultracentrifugation (AUC) of the protein complex used for crystallization and the complex of the full-length SPX2$^{1–280}$/PHR2$^{225–362}$ (no mH2A1.1$^{181–366}$ tag and no internal-residues deletion), respectively (Supplementary Fig. 5). We found that either the protein complex used for crystallization or the complex of full-length SPX2$^{1–280}$ and PHR2$^{225–362}$ are both assembled by a 2:2 stoichiometry ratio in solution (Supplementary Fig. 5), consistent with the crystal structure. This indicate that the mH2A1.1$^{181–366}$ fusion tag and internal-residues deletion of SPX2 have little impact on the SPX2 structure and its ability to bind PHR2. To further validate the crystal structure, we performed small-angle X-ray scattering (SAXS) analysis. We collected SAXS data of the crystallized construct, the

complex of mH2A1.1$^{181-366}$-tagged SPX2$^{1-202/\Delta47-59}$ and PHR2$^{225-362}$. SAXS experiments performed at three different concentrations yielded very similar scattering profiles (Supplementary Fig. 6a). This indicates that the association between mH2A1.1$^{181-366}$-tagged SPX2$^{1-202/\Delta47-59}$ and PHR2$^{225-362}$ is concentration-independent, and that little protein aggregation occurs. However, the experimental scattering data is significant different from the theoretical scattering profile of the crystal structure (Supplementary Fig. 6a), possibly because we omitted disordered residues in our crystal complex structure from these calculations. Furthermore, the fused mH2A1.1$^{181-366}$ tag may be oriented differently in solution when compared to its orientation in the crystal lattice, leading to a discrepancy between the theoretical scattering profile and experimental data. We thus removed the mH2A1.1$^{181-366}$ tag and used the construct of SPX2$^{1-202}$/PHR2$^{225-362}$ to evaluate just these parts of SPX2$^{1-202}$ and PHR2$^{225-362}$ in the mH2A1.1$^{181-366}$-tagged crystal structure in additional SAXS experiments (Supplementary Fig. 6b, and Methods). We found that the theoretical SAXS profile derived from the crystal structure is consistent with that observed in solution (Supplementary Fig. 6b), indicating that the crystal structure is maintained in solution and that the internal-residues deletion and mH2A1.1$^{181-366}$ fusion of SPX2 have little impact on its structure and PHR2 association. Taken together, the AUC and SAXS results corroborate the crystal structure of the SPX2$^{1-202/\Delta47-59}$/InsP$_6$/PHR2$^{225-362}$ complex.

In the SPX2/InsP$_6$/PHR2 complex, the SPX2 dimer obtains a domain-swapped conformation, in which the N-terminal domain of one SPX2 protomer interacts with the C-terminal domain of the other SPX2 protomer, forming an intertwined dimer (Fig. 1b). Specifically, the domain swap involves helices α1 and α2 (NTD) from one protomer and helices α4 and α5 (CTD) from another protomer, which are bridged by two antiparallel extended helices α3 from these two protomers. We validated this domain-swapped dimeric conformation in solution by thiol-directed chemical crosslinking (Supplementary Fig. 7). For this validation, we introduced a cysteine mutation into SPX2 (K106C), which is spatially adjacent to the endogenous cysteine (C182) in another protomer (Supplementary Fig. 7a). The distance between K106C and C182 in the SPX2 dimer would allow them to be cross-linked by 1,2-Ethanediyl Bismethanethiosulfonate (M2M). Consistent with this, the SPX2 in the SPX2/InsP$_6$/PHR2 complex was cross-linked by migrating with higher mass than SPX2 monomer in the SDS-PAGE gel, and the cross-linked SPX2 can be reversibly reduced by dithiothreitol (DTT) (Supplementary Fig. 7b). Neither the wild-type SPX2, possesses the endogenous C182, nor the K106C/C182S double mutant can be cross-linked (Supplementary Fig. 7b). These data indicate that the crosslinking between K106C and C182 is specific, and the domain-swapped conformation of SPX2 dimer in the SPX2/InsP$_6$/PHR2 complex is maintained in solution.

To further validate the domain-swapped SPX2 dimer in the SPX2/InsP$_6$/PHR2 complex and to confirm that the crosslinking occurred between two distinct SPX2 molecules, we encoded the K106C and C182S mutation on two separate SPX2 for chemical crosslinking (Supplementary Fig. 8). We co-expressed SPX2, flag-tagged SPX2 and His-tagged PHR2 in *E.coli* (Supplementary Fig. 8a, and Methods). In the presence of InsP$_6$, three types of SPX2/InsP$_6$/PHR2 complex should be formed, including the PHR2 in complex with two kinds of homo SPX2 dimers (SPX2/SPX2 dimer and SPX2-flag/SPX2-flag dimer) and a hetero SPX2 dimer (SPX2-flag/SPX2 dimer). Following by Ni-NTA pull-down and flag pull-down, PHR2 in complex with SPX2-flag/SPX2-flag homodimer and SPX2-flag/SPX2 heterodimer would be co-eluted (Supplementary Fig. 8a). Using flag pull-down, the SPX2 was co-eluted with SPX2-flag and PHR2-His (Supplementary Fig. 8b, line 2 and 3). This indicates that the SPX2 dimerizes with SPX2-flag to bind PHR2-His. By introducing K106C and C182S mutation on the two separate SPX2 protomers (SPX2 and SPX2-flag), we performed crosslinking experiments on the mixture of SPX2-flag/SPX2-flag homodimer and SPX2-flag/SPX2 heterodimer (Supplementary Fig. 8b). Consistent with the domain-swapped conformation of SPX2 dimer, neither crosslinking occurred within SPX2-flag/SPX2-flag homodimer and SPX2-flag/SPX2 heterodimer (Supplementary Fig. 8b, line 2 and 3), nor within SPX2$^{C182S}$-flag/SPX2$^{C182S}$-flag homodimer and SPX2-flag$^{C182S}$/SPX2$^{K106C}$ heterodimer (Supplementary Fig. 8b, line 4 and 5). In addition, the crosslinking occurred predictably within the SPX2-flag/SPX2$^{K106C}$ heterodimer (Supplementary Fig. 8b, line 6 and 7), or within the SPX2$^{K106C}$-flag/SPX2$^{K106C}$-flag homodimer (Supplementary Fig. 8b, line 10 and 11). And, the crosslinking was mostly occurred within both the SPX2$^{K106C}$-flag/SPX2$^{K106C}$-flag homodimer and SPX2$^{K106C}$-flag/SPX2 heterodimer (Supplementary Fig. 8b, line 8 and 9). Collectively, these results validate the domain-swapped conformation of SPX2 dimer in the complex of SPX2/InsP$_6$/PHR2.

Rice SPX2 is a stand-alone SPX protein, and a 3-dimensional structural homology search with the program DALI[29] revealed that its dimeric conformation has not been previously observed with other SPX domain structures[23,30] (Fig. 1c, d). SPX domain-containing proteins for which ligand-bound structures are available, such as *Chaetomium Thermophilum* glycerophospho-diester phosphodiesterase 1 (SPX$^{CtGde1}$) or Vacuolar transporter chaperone 4 (SPX$^{CtVtc4}$), InsP$_6$ binds the monomeric SPX domain in a 1:1 stoichiometry ratio (Fig. 1c). In these previous structures, core helices α3 and α4 bridge helices α1, α2, α5 and α6, stabilizing a monomeric fold, and InsP$_6$ mainly interacts with helices α2 and α4. In contrast, in our rice SPX2/InsP$_6$/PHR2 complex, SPX2 adopts a domain-swapped dimer conformation and coordinates two InsP$_6$ molecules in a 2:2 stoichiometry ratio. The helix α2 of one protomer and the helix α3 of another protomer form the basic binding surface for InsP$_6$ / PP-InsPs.

**Recognition of InsP$_6$ by the domain-swapped SPX2 dimer in the SPX2/InsP$_6$/PHR2 complex.** To assess the recognition of inositol pyrophosphate signal by the domain-swapped rice SPX2 dimer, we performed extensive mutational analyses of the InsP$_6$ / PP-InsP binding site. We co-expressed 8×His-tagged PHR2$^{225-362}$ and untagged wild-type vs. mutant full-length SPX2 (comprising residues 1-280) and assessed their interaction in Ni-NTA pull-downs in vitro (see Methods). InsP$_6$ binds to a positively charged surface by a set of highly conserved residues in SPX proteins[23] (Fig. 2a, b; Supplementary Figs. 3 and 9). The binding surface in rice SPX2 is formed by the basic residues K26, K29 and R31 from protomer A, and K143, K146, K147 and K150 from protomer B of the SPX2 dimer. The highly conserved Y25 and L28 also contribute the binding surface. Consequently, the InsP$_6$-binding dependent SPX2 – PHR2 association was strongly reduced or abolished by the Y25F, Y25A, L28A, K29A, or K143A/K147A substitutions of SPX2 (Fig. 2c), indicating that these mutations have perturbed the sensing of InsP$_6$ by SPX2. Single amino-acid substitutions in the SPX basic binding surface had little effect on SPX2 – PHR2 association (Fig. 2c). The area of the positively charged accessible surface for InsP$_6$ binding is larger than the shape of negatively charged InsP$_6$ (Fig. 2b), suggesting that, like previously shown for other SPX domains, PP-InsPs such as InsP$_7$ or InsP$_8$, may represent the bona fide Pi signalling molecule recognized / sensed by rice SPX2[23,27,31].

Our crystal structure and solution-based structural characterization reveal a domain-swapped SPX2 dimer in the rice SPX2/

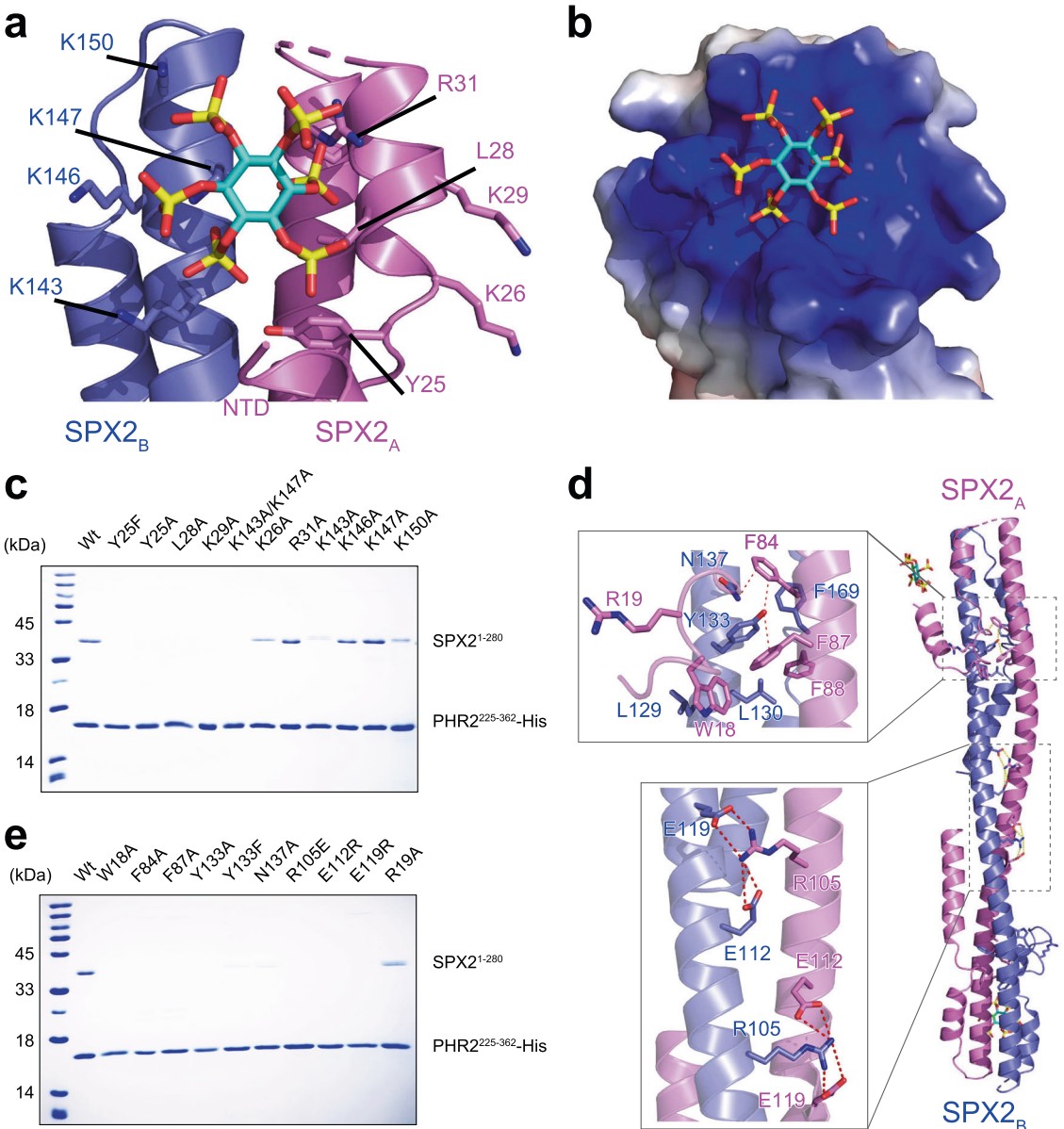

**Fig. 2 The recognition of InsP$_6$ by the domain-swapped rice SPX2 dimer in the SPX2/InsP$_6$/PHR2 complex. a** Binding surface of InsP$_6$ in the SPX2 dimer. **b** The electrostatic surface of the InsP$_6$ binding surface, colored in terms of electrostatic potential, and displayed in a scale from red (−5 kT/e) to blue (+5 kT/e). **c** Co-expression coupled Ni-NTA pull-down assess the InsP$_6$-binding dependent SPX2–PHR2 association. Experiments were independently repeated three times with similar results. **d** Interface of the SPX2 dimer. PHR2 has been omitted for clearity. **e** Co-expression coupled Ni-NTA pull-down assess the SPX2 dimerization for SPX2/PHR2 association. Experiments were independently repeated three times with similar results. For the pull-down assay, different mutated versions of the full length SPX2$^{1-280}$ and His-tagged PHR2$^{225-362}$ were co-expressed in the presence of 1 mM InsP$_6$.

InsP$_6$/PHR2 complex, we next assessed the role of the SPX2 dimer interface in PHR2 binding. The intertwined dimer is mainly stabilized by hydrophobic and paired electrostatic interactions between the two anti-parallel helices α3. The hydrophobic network encompasses W18, F84, F87 and F88 from one protomer, and L129, L130, Y133 and N137 from another protomer (Fig. 2d). Substitutions of key residues in the network disrupted the PHR2 – SPX2 complex in vitro, whereas a substitution outside the hydrophobic network, R19A, had little impact on PHR2 – SPX2 complex association (Fig. 2d, e). The paired electrostatic interactions involve R105, E112 and E119 from one protomer, and E119, E112 and R105 from another protomer (Fig. 2d). Charge reversal mutation of these residues (R105E, E112R, and E119R) in the SPX2 dimer, that breaks paired electrostatic interactions, also abolished the binding of

PHR2 (Fig. 2d, e). These residues contributing to SPX2 dimerization are well conserved in the stand-alone SPX proteins (Supplementary Fig. 3), supporting their importance in the dimerization of SPX proteins for the binding of PHR transcription factors.

**Structural basis for PHR2 targeting and mechanism for PHR2 inactivation by PP-InsPs-bound SPX2.** We next assessed the interaction interface between SPX2 and PHR2 in vitro pull-down assays. Two copies of PHR2 are wrapped around the domain-swapped SPX2 dimer (Fig. 3a). The MYB domain and CC domain of PHR2 mainly interacts with the helix α3 of protomer A, and helices α3 and α5 of the protomer B in the SPX2 dimer, respectively. Residues R250, E257, H294, K297, Y298 and R302 in the

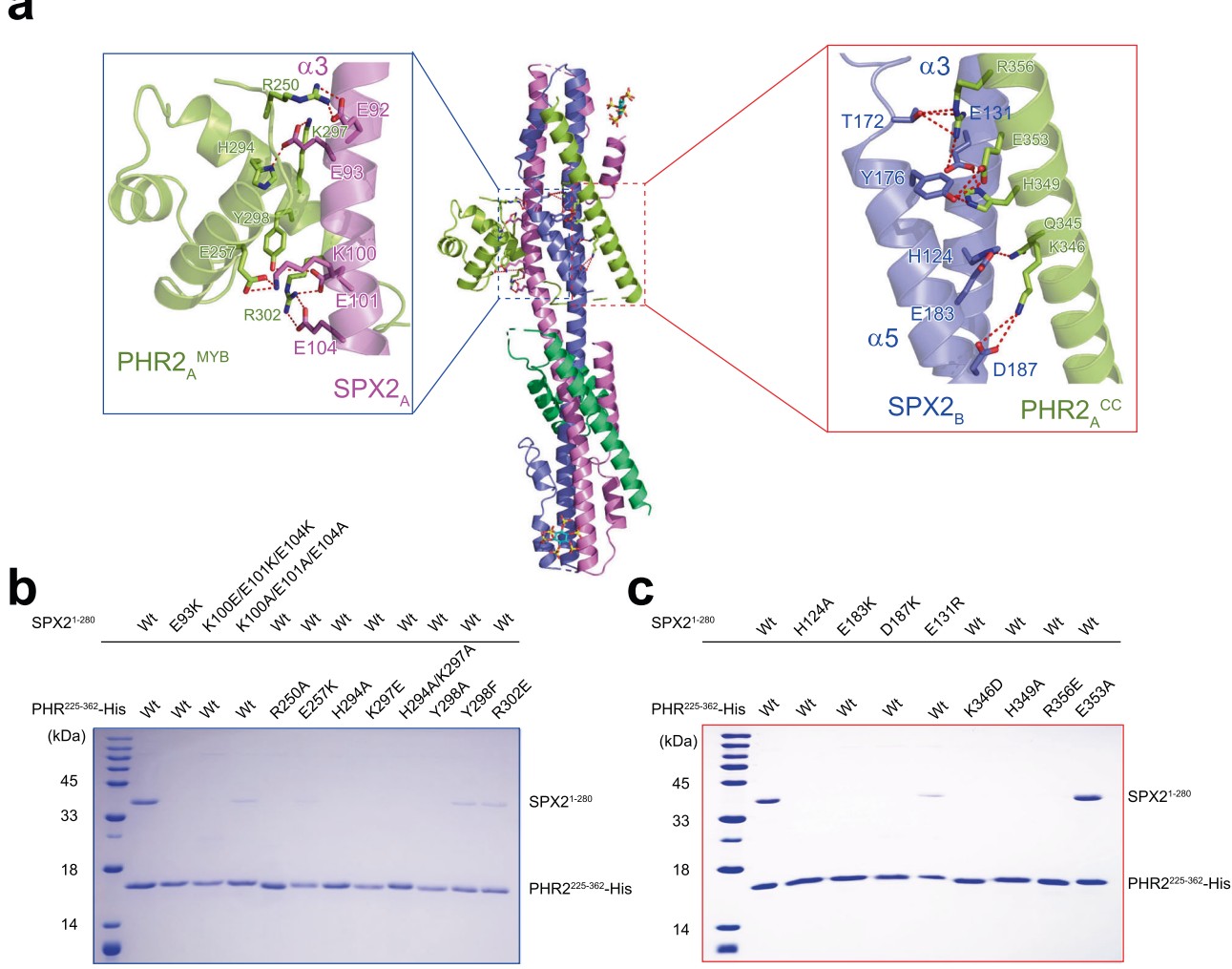

**Fig. 3 The recognition of rice PHR2 by the domain-swapped rice SPX2 dimer. a** The binding surface of the MYB domain and CC domain of $PHR2_A$ in the domain-swapped SPX2 dimer. The PHR2 protomer A binds to the SPX2 dimer in the same way as protomer B, and only the details of protomer A are depicted. **b** Co-expression coupled Ni-NTA pull-down assess the interface between $PHR2^{MYB}$ and SPX2 dimer, and **c** between $PHR2^{CC}$ and SPX2 dimer. Experiments were independently repeated three times with similar results. For the pull-down assay, different mutated versions of the full length $SPX2^{1-280}$ and His-tagged $PHR2^{225-362}$ were co-expressed in the presence of 1 mM $InsP_6$.

MYB domain, and E92, E93, K100, E101 and E104 in the helix α3 of SPX2 protomer A, mediate the $PHR2^{MYB}$ – SPX2 association. The interaction between the CC domain and SPX2 is stabilized by residues Q345, K346, H349, E353 and R356 in the CC domain and H124, E131, T172, Y176, E183 and D187 in the helices α3 and α5 of SPX2 protomer B (Fig. 3a). Consistent with this, mutations of these key interacting residues in the $PHR2^{MYB}$ – SPX2 interface or in the $PHR2^{CC}$ – SPX2 interface disrupted the association of SPX2 and PHR2 (Fig. 3b, c). We also validated these interactions by in vitro pull-down analysis using rice SPX4, a homolog of rice SPX2 (Supplementary Fig. 10).

To gain insights into how rice SPX2 reduces the DNA binding ability of PHR2, we determined the structure of the PHR2 MYB domain in complex with P1BS motif at 2.7 Å resolution (Fig. 4a and Supplementary Table 1). It turns out that two MYB molecules are coordinated into the major groove of the imperfect palindromic DNA mainly through helices α3. The P1BS recognition by MYB is mediated by the interaction between the residues K292, S293 and Q296 in MYB and the nucleotides G7, A9, A11, G4′, A6′ and A8′ in P1BS (Fig. 4b). Electrophoretic mobility shift assay (EMSA) results revealed that alanine substitution of theses residues attenuated the DNA binding

ability of PHR2, and the charge reversal mutation of K292 abolished this interaction (Supplementary Fig. 11a). The structure of rice MYB/P1BS complex is consistent with classical binding models[32], and is similar to the reported structure of the *Arabidopsis* PHR1 (AtPHR1) MYB in complex with P1BS (Supplementary Fig. 11b)[25].

The structures of $SPX2/InsP_6/PHR2$ complex and MYB/DNA complex provide a framework for understanding how SPX2 senses the PP-InsP signal and transduces this signal into transcription inhibition. Superposing the two complex structures using the MYB domains as a reference, the position of SPX2 molecules heavily overlapped with the DNA position in the MYB/DNA complex, indicating that the binding of $InsP_6$-bound SPX2 to PHR2 would produce severe steric clashes and thus preventing DNA binding (Fig. 4c). It has been previously established that the CC domain of PHRs enables oligomerisation of the transcription factor critical for DNA binding[17,26]. In line with this, EMSA results showed that the DNA binding ability of $PHR2^{MYB-CC}$ is stronger than $PHR^{MYB}$ (Supplementary Fig. 11c). Our recently study indicated that mutation of K325, H328 and R335 at the surface of *Arabidopsis* PHR1 CC domain disrupted its interaction with *Arabidopsis* SPX receptors, and led

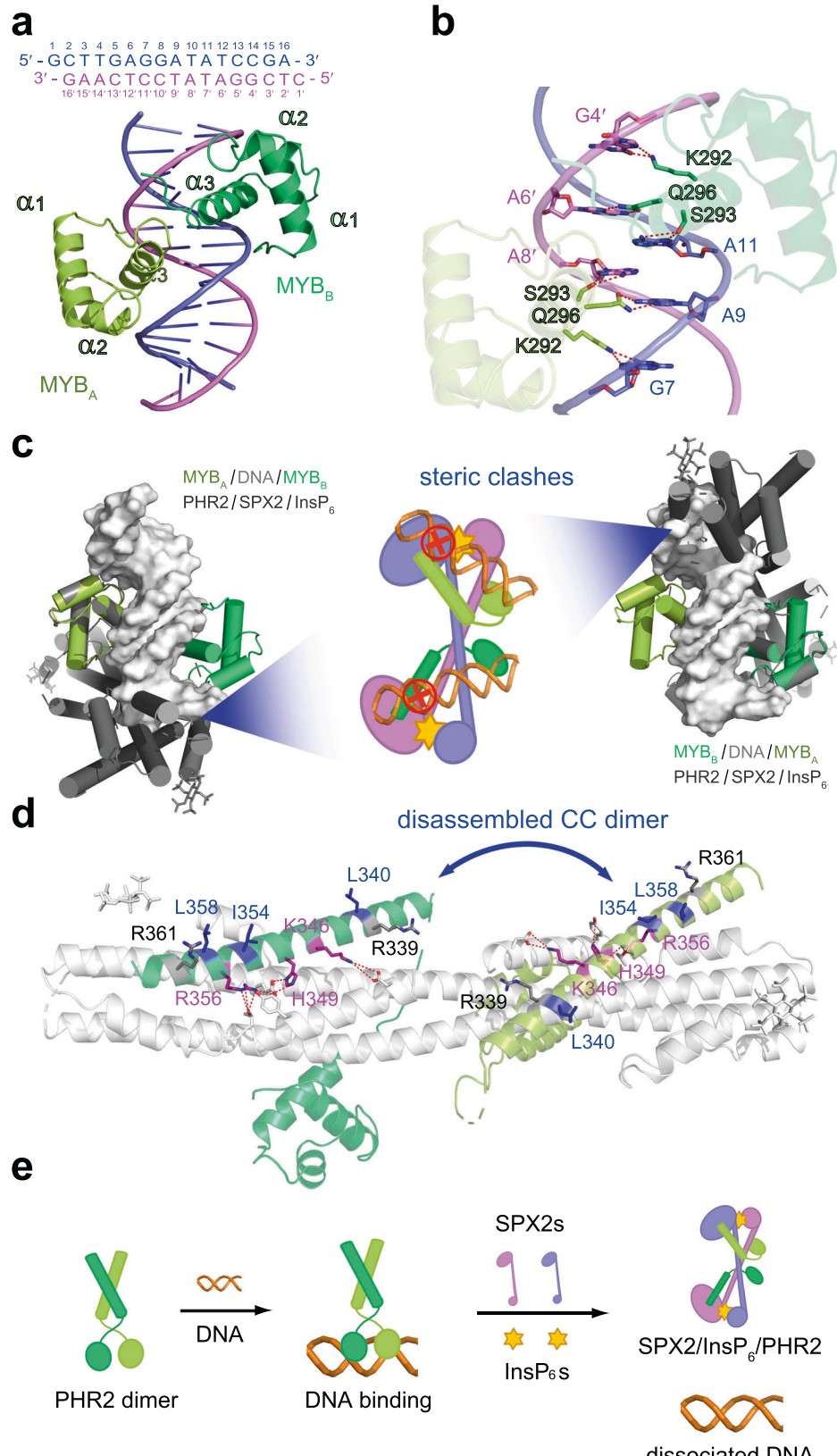

to constitutive Pi starvation responses in *Arabidopsis*[26], whereas the mutation of R318 or R340 produced no effect[26]. Mechanistically, our structure shows that these homologous residues K346, H349 and R356 in OsPHR2 (corresponding to K325, H328, and R335 in AtPHR1) interact with OsSPX2, whereas R339 and R361 (corresponding to R318 and R340 in AtPHR1)

do not contribute to the formation of the signalling complex (Fig. 3a, c and Fig. 4d).

We have previously reported that the residues L319, I333 and L337 in AtPHR1 stabilize the CC oligomer required for DNA/promoter binding[26]. Our complex structure now uncovers the mechanism by which rice SPX2 disassembles PHR2 CC oligomers

**Fig. 4 The mechanism of transcriptional repression of rice PHR2 by the InsP$_6$-bound SPX2 binding. a** The structure of PHR2 MYB domain in complex with DNA. The P1BS motif is highlighted in gray shading. **b** The interface between the MYB domain and P1BS motif. **c** Structure superposition of MYB/DNA complex and SPX2/InsP$_6$/PHR2 complex. They are aligned by superposing the MYB protomer A or B of MYB/DNA structure with the MYB domain of PHR2 protomer A or B in the SPX2/InsP$_6$/PHR2 complex, respectively. The two MYB molecules and DNA in the MYB/DNA complex are colored in yellow cartoon, green cartoon and gray surface representation, respectively. Steric clashes of PHR2 preventing DNA binding are illustrated in the middle model. **d** Structural basis for InsP$_6$-bound SPX2 disassembles the dimerization of PHR2 CC domain. Residues responsible for SPX2 binding and CC dimerization are highlighted in magenta and blue, respectively. No-interacting residues are colored in black. **e** A model illustration for the transcription inhibition of PHR2 by InsP$_6$/PP-InsPs.

and impairs DNA binding. Specifically, SPX2 blocks the assembly of PHR2$^{CC}$ oligomers by exposing hydrophobic residues normally contributing to the stabilization of the coiled-coil structure (including L340, I354 and L358 that correspond to the previously characterized L319, I333 and L337 in AtPHR1)[26] (Fig. 4d). Collectively, by forming the SPX2/InsP$_6$/PHR2 complex, a domain-swapped SPX2 dimer senses InsP$_6$/PP-InsP nutrient messengers to repress the PHR2 transcriptional activity by two distinct mechanisms: 1) The association of the InsP$_6$-bound SPX2 establishes severe steric clashes with the MYB domain, preventing DNA binding 2) Disruption of the CC domain oligomerisation motif in PHR2 attenuates its transcriptional activity (Fig. 4c, d).

## Discussion

The dynamic interactions between PP-InsP nutrient messengers, SPX receptor proteins, PHR family transcription factors and corresponding PSI genes are the master regulation network in plant Pi homeostasis maintenance[1,2], and also in plant – microorganism communication[19]. Here we have determined the crystal structures of the PP-InsPs-bound rice SPX2 in complex with PHR2 and of the PHR2 MYB domain in complex with its target DNA. In the SPX2/InsP$_6$/PHR2 complex, the signalling-active SPX2 assembles into a domain-swapped dimer conformation to bind two copies of PHR2. Although the binding surface of PP-InsP is highly conserved in SPX receptor proteins[23] (Fig. 2a, b; Supplementary Fig. 3), the rice SPX2 adopts a different topology to recognize PP-InsP compared to other reported SPX receptor proteins (Fig. 1). Our AUC, SAXS and chemical crosslinking results have collectively conformed that the domain-swapped conformation of SPX2 dimer in the complex of SPX2/InsP$_6$/PHR2 is maintained in solution (Supplementary Figs. 5–8).

While our manuscript was under review, Zhou and colleagues reported the crystal structure of rice SPX1$^{1–198}$ in complex with PHR2$^{248–380}$ and InsP$_6$[33]. Interestingly, this complex structure reveals a SPX1 monomer that binds one molecule of InsP$_6$ and a PHR2 monomer with 1:1 stoichiometry (Supplementary Fig. 12). In contrast, our complex structure revealed an InsP$_6$-bound SPX2 domain-swapped dimer that binds PHR2 with a 2:2 stoichiometry ratio (Supplementary Fig. 12). To clarify if the dimerization of SPX2 is induced by the binding of InsP$_6$, we sought to use analytical size-exclusion chromatorgrapy (SEC) and anatlytical ultracentrifugation (AUC) assays. After numerous attempts to obtain soluble SPX2 in the absence of the macro fusion tag, we could purify SPX2$^{1-267/Δ47-64}$ in sufficient quantities to perform these experiments (see Methods). SEC assays revealed that SPX2$^{1-267/Δ47-64}$ exists in different conformations/oligomeric states in the absence of InsP$_6$. In the presence of InsP$_6$, SPX2$^{1-267/Δ47-64}$ appears to behave as a monomer in SEC assays (Supplementary Fig. 13a, b). Consistently, AUC analysis revealed inhomogeneous species of SPX2$^{1-267/Δ47-64}$ in the absence of InsP$_6$, with a major species of ~33.2 kDa in molecular weight (Supplementary Fig. 13c). In the presence of InsP$_6$, SPX2$^{1-267/Δ47-64}$ behaves as a single species with a molecular weight of ~27.8 kDa (Supplementary Fig. 13c), in good agreement with a SPX2 monomer

(theoretical molecular weight of SPX2$^{1-267/Δ47-64}$ is ~29.3 kDa). Thus, in the absence of PHR2, InsP$_6$ binding/sensing alone cannot induce SPX2 dimerization. The domain-swapped dimer conformation of SPX2 in our SPX2/InsP$_6$/PHR2 complex structure may be contributed by PHR2 binding. The broad SEC profile and inhomogeneous AUC species of apo SPX2$^{1-267/Δ47-64}$ however strongly suggest, that SPX2 can exist in different conformations. InsP$_6$ binding appears to stabilize the domain into a monomeric state. This is consistent with the reported results in which InsP$_6$ binding stabilizes the α1 of rice SPX1[33].

To clarify if the InsP$_6$-induced changes are shared by other SPX proteins, we sought to further analyze other the full-length standalone SPX proteins from rice and *Arabidopsis* (including OsSPX1, OsSXP2, OsSPX3, OsSPX4, OsSPX5, OsSPX6, AtSPX1, AtSPX2, AtSPX3 and AtSPX4). We could obtain soluble AtSPX2, AtSPX4, OsSPX1 and OsSPX4. Importantly, their SEC profiles differ strongly in the presence of InsP$_6$ (Supplementary Fig. 14), suggesting that different plant SPX receptors may undergo different conformational changes upon PP-InsPs sensing. Further comprehensive structural studies of these SPX proteins in their apo and PP-InsPs-bound form will be needed to assess the functional significance of the different conformations and oligomeric states.

Altogether, our structural characterizations and biochemical analysis allow us to propose a molecular model for the transcription inhibition of PHR2 by InsP$_6$ / PP-InsPs (Fig. 4e). Our structure clearly defines that in the case of PHR – SPX interactions, PP-InsPs do not act as "intermolecular glue" promoting the association of the signalling complex[34]. Instead, in the rice SPX2/InsP$_6$/PHR2 complex, the signalling-active SPX2 assembles into a domain-swapped dimer conformation that targets two PHR2 monomers forming a 2:2:2 complex. Upon the binding of signalling-active SPX2, the PHR2 CC domain cannot longer oligomerise and the PHR2 MYB domain is inaccessible to DNA binding, leading to PHR2 inactivation. Our work and reported studies[26,33] collectively uncover the molecular mechanism of plant phosphate homeostasis and provide a framework for the rational engineering of crops with improved Pi use efficiency.

## Methods

**Molecular cloning**. The codon-optimized complementary DNA of full length *SPXs* and *PHR2* were synthesized. Gibson Assembly method was used for cloning constructing. *PHR2* was subcloned into a pET21B (Novagen) vector with a C-terminal 8×His tag, using restriction endonucleases NdeI and XhoI. *OsSPX1, OsSPX4, AtSPX2 and AtSPX4* was constructed with a 6×His tag and a caspase drICE protease cleavage site at the N-terminus, and was subcloned into a pET15D vector using restriction endonucleases NdeI and XhoI, respectively. *OsSPX2* was cloned into a pBB75 vector without any tag using restriction endonucleases NdeI and EcoRI. The site-specific mutations were introduced into *PHR2* or *SPX* genes by overlap PCR. All constructs were verified by DNA sequencing. Sequences of all relevant oligonucleotide primers are summarized in Supplementary Table 2.

**Protein expression and purification**. For the preparation of rice SPX2/PHR2 complex, the SPX2 and PHR2, or the particular boundary and mutants, were co-expressed in *E. coli* strain BL21(DE3) using Lysogeny broth (LB) medium. The cells were induced with 0.2 mM isopropyl-β-D-thiogalactoside (IPTG) and 1 mM InsP$_6$ at 16 °C for 12 h. Harvested cells were lysed by a high-pressure cell disrupter in a buffer containing 25 mM Tris–HCl pH 8.0,150 mM NaCl, 1 mM InsP$_6$. Target protein was collected from the supernatant and purified over Ni$^{2+}$ affinity resin

and HiTrap Heparin column used in tandem. 1 mM InsP$_6$ was present during all the purification processes. The protein was further purified into homogeneity by gel-filtration chromatography (Superdex-200 Increase 10/300 GL, GE Healthcare) in a buffer containing 25 mM Tris–HCl pH 8.0, 150 mM NaCl, 5 mM DTT and 1 mM InsP$_6$. Target fractions were collected for biochemistry experiments and supplied with 10 mM InsP$_6$ for crystallization.

PHR2 was expressed alone and purified similarly as the purification of rice SPX2/PHR2 complex, except there was no InsP$_6$ added.

OsSPX1, OsSPX4, AtSPX2 and AtSPX4 was expressed in *E. coli* strain BL21(DE3) using LB medium, and induced with 0.2 mM IPTG at 16 °C for 12 h, respectively. Harvested cells were lysed, and the target protein was purified over Ni$^{2+}$ affinity resin. After removal of the His tag by drICE protease, target protein was further purified over Source 15Q and Superdex-200 Increase 10/300 columns used in tandem. For the pull-down assay of OsSPX4 and OsPHR2, OsSPX4 was finally prepared in a buffer containing 25 mM Tris-HCl pH 8.0, 150 mM NaCl and 1 mM InsP$_6$. For the analytical size-exclusion chromatorgrapy (SEC) assays, each SPX protein was finally prepared in a buffer containing 25 mM Tris-HCl pH 8.0, 150 mM NaCl.

For the preparation of OsSPX2$^{1-267/\Delta47-64}$, the gene was constructed in pBB75 vector with a C-terminus 8×His tag. OsSPX2$^{1-267/\Delta47-64}$ was co-expressed with OsPHR2$^{231-362}$ (constructed in pET21b vector without C-terminus His tag). The cells were induced with 0.2 mM isopropyl-β-D-thiogalactoside (IPTG) and 1 mM InsP$_6$ at 16 °C for 12 h. Harvested cells were lysed, and the target protein was purified over Ni$^{2+}$ affinity resin. The elution was subjected onto HiTrap Heparin column and the excess SPX2$^{1-267/\Delta47-64}$ was separated from the SPX2$^{1-267/\Delta47-64}$/PHR2$^{231-362}$ complex. Theses excess SPX2$^{1-267/\Delta47-64}$ was collected and subjected for further purification. For the Ni$^{2+}$ affinity and Heparin purification, 1 mM InsP$_6$ was present during all the purification processes. Collected excess OsSPX2$^{1-267/\Delta47-64}$ was further purified over Source 15Q and Superdex-200 Increase 10/300 columns used in tandem, in the absence of InsP$_6$. OsSPX2$^{1-267/\Delta47-64}$ was finally prepared in a buffer containing 25 mM Tris-HCl pH 8.0, 150 mM NaCl.

MYB domain of PHR2 (residues 180-313) was expressed in *E. coli* strain BL21(DE3) using LB medium, and induced with 0.2 mM IPTG at 16 °C for 16 h. Harvested cells were lysed, and the target protein was purified over Ni$^{2+}$ affinity resin, Source 15Q, and Superdex-200 Increase 10/300 columns used in tandem. The protein was final prepared in a buffer containing 25 mM Tris-HCl pH 8.0, 150 mM NaCl, 5 mM MgCl$_2$, and 5 mM DTT. For crystallization trials, the PHR2 MYB was incubated with the P1BS (5′-gCTTGAGGATATCCGA-3′ and 5′-cTCGGATATCCTCAAG-3′) in a molar ratio of 1:1.5 at 4 °C for about 30 min.

### Crystallization

Crystallization experiments were performed through hanging-drop vapour-diffusion methods by mixing the protein with an equal volume of reservoir solution at 18 °C. The mH2A1.1$^{181-366}$ tagged SPX2$^{1-202/\Delta47-59}$/InsP$_6$/PHR2$^{225-362}$ complex gave rise to best crystals under the condition of 8.4% PEG5000 MME, 6.5% Tacsimate pH 7.0, 0.15 M (NH$_4$)$_2$SO$_4$, 0.1 M HEPES pH 6.8, 2.67% Pentaerythritol ethoxylate (3/4 EO/OH). Crystals were equilibrated in a cryoprotectant buffer containing 8%-12% PEG5000 MME, 5% Tacsimate pH 7.0, 0.1 M (NH$_4$)$_2$SO$_4$, 0.1 M HEPES pH 7.0, 0.1 M KCl, 1.5 M L-Proline. By stepwise dehydration in the air, the crystal quality was improved. Finally, a best crystal from thousands of dehydrated crystals diffracted to 3.1 Å at Shanghai Synchrotron Radiation Facility beamline BL17U.

To crystallize the MYB domain of rice PHR2 (residues 180-313) in complex with DNA, the purified MYB was concentrated to about 200 μM and incubated with double-strand DNA (5′-gCTTGAGGATATCCGA-3′ and 5′-cTCGGATATCCTCAAG-3′). The crystal of MYB/DNA complex was grown in 13% (w/v) PEG 3350, 0.1 M sodium malonate pH 6.0. The crystals were flash frozen in liquid nitrogen using 25% glycerol as the cryoprotective buffer and diffracted to 2.7 Å.

### Data collection and structure determination

X-ray diffraction datasets of mH2A1.1$^{181-366}$-tagged SPX2$^{1-202/\Delta47-59}$/InsP$_6$/PHR2$^{225-362}$ complex and MYB/DNA complex were collected at the Shanghai Synchrotron Radiation Facility (SSRF) on beamline BL17U or BL19U[35,36]. The data were integrated and processed with the HKL2000 program suite or in XDS package[37]. Further data processing was carried out using CCP4 suit[38]. Crystal structures of the mH2A1.1$^{181-366}$-tagged SPX2$^{1-202/\Delta47-59}$/InsP$_6$/PHR2$^{225-362}$ complex and MYB/DNA complex were determined at resolutions of 3.1 Å, 2.7 Å, respectively. The structure of MYB/DNA was solved by molecular replacement using MYB domain of AtPHR1 (PDB ID 6J4R) as a search template. Using the resolved MYB domain and the human histone mH2A1.1 (PDB ID 1ZR3) as search the models, we determined the structure of mH2A1.1$^{181-366}$-tagged SPX2$^{1-202/\Delta47-59}$/InsP$_6$/PHR2$^{225-362}$ complex through molecular replacement by the program PHASER[39]. All the structures were iteratively built with COOT[40] and refined using PHENIX program[41]. Data collection and structure refinement statistics were summarized in Supplementary Table 1. All figures were generated using the PyMOL program (http://www.pymol.org/).

### Analytical ultracentrifugation (AUC)

The AUC experiment was performed in a Beckman Coulter XL-I analytical ultracentrifuge using two-channel centerpieces. Complex of SPX2$^{1-280}$/PHR2$^{225-362}$ and mH2A1.1$^{181-366}$-tagged SPX2$^{1-202/\Delta47-59}$/

PHR2$^{225-362}$ was prepared in a solution of 25 mM Tris-HCl pH8.0, 1 mM InsP$_6$, and 150 mM NaCl, respectively. OsSPX2$^{1-267/\Delta47-64}$ was prepared in a solution of 25 mM Tris-HCl pH8.0, 150 mM NaCl and with or without 1 mM InsP$_6$. Data was collected via absorbance detection at 18 °C for protein at a concentration of 0.7 mg ml$^{-1}$ and rotor speed of 147,420 g. The SV-AUC data were globally analyzed using the SEDFIT program and fitted to a continuous c(s) distribution model to determine the molecular weight.

### Small angle X-ray scattering (SAXS)

SAXS data were collected at the BL19U2 beamline of the Shanghai Synchrotron Radiation Facility (SSRF) at room temperature. 284 μM, 142 μM and 71 μM mH2A1.1$^{181-366}$-tagged SPX2$^{1-202/\Delta47-59}$/PHR2$^{225-362}$ complex, and 30 μM SPX2$^{1-202}$/PHR2$^{225-362}$ complex was prepared in the buffer of 25 mM Tris–HCl pH 8.0, 150 mM NaCl, 5 mM DTT and 1 mM InsP$_6$ for SAXS measurement, respectively. For each measurement, 20 consecutive frames of 1-sec exposure were recorded and averaged, providing no difference between the first and the last frames. The background scattering was recorded for the matching buffer and was subtracted from the protein scattering data. The SAXS experiment was performed at room temperature. The data was visualized and analyzed using the software package ATSAS[42]. The theoretical SAXS profile of mH2A1.1$^{181-366}$-tagged SPX2$^{1-202/\Delta47-59}$/PHR2$^{225-362}$ crystal structure was calculated using CRYSOL in ATSAS package. For the calculation of the theoretical SAXS profile of SPX2$^{1-202}$/InsP$_6$/PHR2$^{225-362}$ complex, the invisible residues (including I35-M66, P191-G202 of SPX2$^{1-202}$ and S225-T247, L307-G329 of PHR2$^{225-362}$) and the deleted internal-residues R47-T59 of SPX2$^{1-202}$ in the crystal structure of SPX2$^{1-202/\Delta47-59}$/InsP$_6$/PHR2$^{225-362}$ complex were patched using PyMOL. Considering the flexibility of these patched residues, this full SPX2$^{1-202}$/InsP$_6$/PHR2$^{225-362}$ structure was subjected to Xplor-NIH[43] for the optimization of these patched residues. During the optimization, the coordinates for the visible crystal parts were fixed, only these patched residues were given full torsional freedom to minimize total energy. 260 conformations were optimized, and each theoretical SAXS profile and radius of gyration (Rg) was calculated using CRYSOL in ATSAS package. The conformation with the lowest energy was used to assess the fitting of the experimental SAXS data.

### Thiol-directed chemical crosslinking

To perform the thiol-directed chemical crosslinking (Supplementary Fig. 7b), the complex of SPX2$^{1-280}$/PHR2$^{225-362}$, SPX2$^{1-280}$_K106C/PHR2$^{225-362}$ and SPX2$^{1-280}$_K106C/C182S/PHR2$^{225-362}$ was prepared in a buffer containing 25 mM Tris–HCl pH 8.0, 150 mM NaCl and 1 mM InsP$_6$, respectively. About 30 μM complex protein was incubated with 200 μM M2M (1,2-ethanediyl bismethanethiosulfonate) at room temperature for 30 min. The reaction mixture was analyzed by SDS-PAGE and Coomassie Blue staining in the condition of 100 mM DTT, or not.

To encode K106C and C182S mutation on two distinct SPX2 molecules for thiol-directed chemical crosslinking (Supplementary Fig. 8b), the SPX2 mutations were firstly introduced using overlap PCR, and then one copy of SPX2 and one copy of SPX2 with a N-terminal 3×flag tag were assembled into a pBB75 vector using separate T7 promoter and T7 terminator. By co-expressing this assembled plasmids with PHR2$^{225-362}$-His (encoded in pET21b) in the presence of 1 mM InsP$_6$, and using Ni-NTA pull-down and flag pull-down in tandem, we got the mixture of PHR2$^{225-362}$-His in complex of SPX2$^{1-280}$-flag/SPX2$^{1-280}$-flag homodimer and SPX2$^{1-280}$/SPX2$^{1-280}$-flag heterodimer for crosslinking. About 30 μM complex protein was incubated with 200 μM M2M at room temperature for 30 min. The reaction mixture was analyzed by SDS-PAGE and Coomassie Blue staining in the condition of 100 mM DTT, or not.

### Co-expression coupled Ni-NTA pull-down

We have tried various expression systems, such as *E. coli.*, *insects*, or *mammalian* cells, to express the full-length rice SPX2 (comprising residues 1-280) alone, but no soluble SPX2 protein was obtained. Therefore, we applied a co-expression coupled Ni-NTA pull-down strategy to indirectly assess the recognition of InsP$_6$ by monitoring the InsP$_6$-binding dependent SPX2 – PHR2 association (Supplementary Fig. 2). In this system, no tag SPX2 was co-expressed with 8 × His-tagged PHR2, and then supernatant of the lysed cells were loaded on Ni-NTA beads. After washing unbound proteins, the His-tagged PHR2 and interacting SPX2 were co-eluted by imidazole.

Since the SPX2 – PHR2 association is InsP$_6$ dependent (Supplementary Fig. 2)[23,27], the loss of InsP$_6$ binding ability will make SPX2 no longer interact with PHR2. Thus, the Ni-NTA pull-down assay of SPX2 – PHR2 association by co-expressing His-tagged PHR2 with SPX2, or SPX2 mutants, in the presence of InsP$_6$, can be used to assess the recognition of InsP$_6$ by SPX2 (Fig. 2c). Similarly, we used this strategy assessed the role of SPX2 domain-swapped dimer in PHR2 recognition (Fig. 2e), and revealed keys residues for the interaction between SPX2 and PHR2 (Fig. 3b, c).

### In vitro pull-down assay of rice SPX4 and PHR2

SPX4 and PHR2 was prepared in a buffer containing 25 mM Tris-HCl pH 8.0, 150 mM NaCl and 1 mM InsP$_6$, respectively. About 20 μM SPX4 was incubated with 20 μM PHR2 on ice for 30 min. Then the mixture was loaded onto Ni-NTA beads and incubated for 30 min. After extensively washing with a buffer containing 25 mM Tris-HCl pH 8.0, 150 mM NaCl, 15 mM imidazole and 1 mM InsP$_6$, the bound protein was eluted by a buffer containing 25 mM Tris-HCl pH 8.0, 150 mM NaCl, 250 mM

imidazole and 1 mM InsP$_6$. The input protein and eluted fractions were analyzed by SDS-PAGE and Coomassie Blue staining.

**Electrophoretic mobility shift assay (EMSA)**. FAM-labeled primers were used to generate the DNA fragment (F:5′-AAGCTTGAATATGCAATGGAATATGCT TAG-3′, R:5′-CTAAGCATATTCCATTGCATATTCAAGCTT-3′). The DNA fragment was annealed by heating to 95 ℃ for 5 min and gradually cooled to 25 ℃. The FAM-labelled DNA (10 nM) was incubated with 0.3375, 0.45, 0.6, and 0.8 µM PHR2 proteins at 4 ℃ for 30 min, in a buffer of 25 mM Tris–HCl pH 8.0, 150 mM NaCl, 5 mM DTT, 10% glycerol, and 200 ng ml$^{-1}$ Heparin. The reactions were resolved on 8% native acrylamide gels (37.5:1 acrylamide:bis-acrylamide) in 0.5× Tris–Boric acid buffer at 150 V for about 3 h. Images of the gels were obtained using FLA5100(Typhoon, Fuji, Japan).

**Reporting summary**. Further information on research design is available in the Nature Research Reporting Summary linked to this article.

## Data availability
Atomic coordinates of rice SPX2/InsP$_6$/PHR2 and PHR2$^{MYB}$/DNA complex have been deposited in the Protein Data Bank (PDB) under accession number 7D3Y and 7D3T, respectively. Source data are provided with this paper.

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

## Acknowledgements
We thank staffs at the BL17U, BL19U and BL19U2 beamline of the NCPSS at Shanghai Synchrotron Radiation Facility for assistance with data collection, and research associate Dr. Delin Zhang at the Center for Protein Research, Huazhong Agricultural University, for facilities support. We thank Prof. Lizhong Xiong for helpful discussions. The work has been supported by the National Key R&D Program of China (2018YFA0507700 to Z.L.), by the National Natural Science Foundation of China (32071226 to Z.L.), by the Foundation of Hubei Hongshan Laboratory (2021hszd011 to Z.L.), by the Fundamental Research Funds for the Central Universities (Program No. 2662019PY004 to Z.L.), by the European Union's Horizon 2020 research program under ERC consolidator grant agreement 818696 (INSPIRE) to M.H., and by the Wuhan Applied Foundational Frontier Project (2019020701011460 to P.Y.). Zeyuan Guan also acknowledges the support of National Postdoctoral Program for Innovative Talents (BX2021108).

## Author contributions
Z.L., and Z.Y.G. conceived the project and designed the experiments, Z.Y.G., Q.X.Z., and Z.F.Z. performed crystallization and resolved the structures, J.S., and L.B., analyzed the structures and designed biochemical experiments, J.Q.Z., J.C., R.W.L., P.C., Q.W., K.P.,

D.L.Z., T.T.Z., J.J.Y., and P.Y. performed biochemistry experiments and analyzed data. Z.L., and M.H. wrote the manuscript with support from all authors.

## Competing interests

The authors declare no competing interests.
