## [Peer Review File · Nature Communications]

Mechanistic insights into the regulation of plant phosphate homeostasis by the rice SPX2 – PHR2 complexREVIEWER COMMENTS

Reviewer #1 (Remarks to the Author):

Guan et al., present two new crystal structures of a 2:2:2 complex of PHR:SPX:IP6 that makes a major contribution to understanding phosphate signaling in plants and how PHR DNA interaction is inhibited by SPX. The work is done very well, is novel and would be interesting to a wide audience. The authors have taken a rigorous approach to validating their model using several solution-based orthogonal approaches which I appreciate, however I have a few concerns:

Somewhat Major:

- 1) No PDB validation report so no comment on the structure until I see it.
- 2) Crosslink data do not distinguish between a) the proposed inter-molecular crosslink between two distinct SPX molecules or b) an intramolecular crosslink. Crosslink mutations in Fig S7b encoded on two separate plasmids/alleles should give similar results if domain-swap model is correct.
- 3) SEC-SAXS of crystallized constructs with all tags and deletions would show domain swap is present in solution. Current SAXS (Fig S6) only examined proteins that were not crystallized, with separate critiques below.
- 4) Structure suggests residues from each protomer stabilize IP6. K147A in protomer B alone does not destabilize, nor does R31A alone in protomer A. If these mutations are epistatic and encoded on separate plasmids/alleles, contribution from each protomer should decrease SPX-PHR binding, testing model.
- 5) Some test of IP6 inducibility should be done. SEC-SAXS in absence of IP6 should be wildly different if tetramer does not form.

Less Major:

- 6) Does alphafold2 predict the extended SPX helix 3, when a monomer (without PHR)?

7) AUC in Fig S5 shows ~10% smaller species which should perturb ensemble SAXS radius of gyration, but the fit up in guinier region looks tight. Guinier fit, chi square and residuals should be shown, and AUC discrepancy addressed in text. As is, looks like something isn't right and there's not enough detail in SAXS provided to know what it is.

8) For calculated SAXS, residues missing from the crystal structure were modeled which can introduce bias as its very easy to put the residues wherever they improve SAXS fit. Authors should make it clear the model was not changed iteratively, and without peaking at the fit with observed SAXS.

9) Data in Figure 2C is inaccurately described in line 110, in Figure 2C and in the methods. Authors state IP6 binding is being measured, and provide their explanation, but they do not measure IP6 binding. They only measure SPX-PHR interaction perturbed by mutations at the putative IP6 site. The mutations are the variable, language should be changed to state this. Or they can actually measure IP6 binding using radiolabels and report Kd's, which would be much better.

Reviewer #2 (Remarks to the Author):

PHR and SPX family proteins are key players in maintaining the homeostasis of Pi. In the presence of inositol pyrophosphates (PP-InsPs), SPX family proteins can interact with PHR proteins, inhibiting their gene activation ability and avoiding the toxicity of high concentration of Pi. In this manuscript, Guan et al. determined the complex structure of rice SPX2/InsP6/PHR2, revealed the detailed interaction between SPX2 and PHR2 and the functional importance of PP-InsPs. In addition, they also determined the crystal structure of PHR2 MYB domain in complex with its target DNA. This work helps to understand the molecular mechanism how SPX2 inhibits PHR2 from binding to the target genes to regulate Pi homeostasis in rice. The major concern is how to interpret that SPX2 formed domain-swapped dimer in the SPX2/InsP6/PHR2 complex.

Concerns:

1) In this work, the authors found that rice SPX2 formed domain-swapped dimer in the ternary complex, which has NOT been previously observed in other SPX domain structures. Although the authors verified the dimerization state of rice SPX2 by Analytical ultracentrifugation (AUC), I still suspect that the dimerization of SPX2 is probably artificial by domain swapping. Please test the dimerization of SPX2 by gel filtration assay to clarify whether the domain swapping is caused by the truncation of PHR2 (225-362). Since the full-length PHR2 is not stable, the authors can check PHR2 (231-426) (Wang et al., PNAS,

2014), full-length SPX2 (1-280), PHR2 (231-426) and SPX2 (1-280) in presence or absence of InsP6 in gel filtration.

If the authors failed to prove that SPX2 could form dimer in gel filtration, their complex structure of rice SPX2/InsP6/PHR2 still makes sense, but they have to clarify that the dimerization state of rice SPX2 in the complex is due to domain swapping.

2) In panel C of Fig. 2, SPX2 K29A mutant failed to interact with PHR2. However, the side chain of SPX2 K29 is pointed away from InsP6 in the structure (see Fig. 2 panel A). Similar problem was also observed for SPX2 K143 in Fig. 2 panels A and C. Please check the conformation of K29 and K143 in the structure. In addition to InsP6, it is also helpful to include the electron density of K29 and other residues critical for InsP6 recognition in the manuscript.

3) In Table S1, please check the B factor for the ternary complex. The overall B-factor is very high (over 147 and 280 Å² for overall structure and other entities). Besides, the B-factor of the side chain atoms is much lower than other atoms.

Reviewer #1 (Remarks to the Author):

Guan et al., present two new crystal structures of a 2:2:2 complex of PHR:SPX:IP6 that makes a major contribution to understanding phosphate signaling in plants and how PHR DNA interaction is inhibited by SPX. The work is done very well, is novel and would be interesting to a wide audience. The authors have taken a rigorous approach to validating their model using several solution-based orthogonal approaches which I appreciate, however I have a few concerns:

Response: We are glad to see that you recognize the importance of our work. We truly appreciate your insightful comments and very constructive suggestions that helped us to significantly improve our manuscript. According to your suggestions, we have performed more experiments and analyses. These results and discussions were included in the revised manuscript. Please find our point-by-point response to your concerns listed below.

Somewhat Major:

1) No PDB validation report so no comment on the structure until I see it.

Response: We are sorry for missing the PDB validation reports. We have uploaded these files in the revision.

2) Crosslink data do not distinguish between a) the proposed inter-molecular crosslink between two distinct SPX molecules or b) an intramolecular crosslink. Crosslink mutations in Fig S7b encoded on two separate plasmids/alleles should give similar results if domain-swap model is correct.

Response: Thank you for your insightful comments and very constructive suggestions. In our previous manuscript, because the cross-linked species of SPX2 migrated with higher mass than SPX2 monomer in the SDS-PAGE, we proposed that the crosslinking was occurred between inter-molecules. We have included more details and discussion in the revised main text (paragraph 2 in page 5 and paragraph 1 in page 6).

Following your constructive suggestion, we encoded the K106C and C182S mutation on two separate SPX2 molecules for crosslinking. We have included these new results in the revised main text (paragraph 2 in page 6 and paragraph 1 in page 7) and added a figure as Supplementary Fig. 8 in the revised Supplementary Information. These new results collectively conformed that the crosslinking was occurred between two distinct SPX2 molecules and validated the domain-swapped conformation of SPX2 dimer in the complex of SPX2/InsP₆/PHR2 in solution.

3) SEC-SAXS of crystallized constructs with all tags and deletions would show domain swap is present in solution. Current SAXS (Fig S6) only examined proteins that were not crystallized, with separate critiques below.

Response: Thank you for your comments and we are happy that you raised this point in which we have tried. We completely agree with you that using the crystallized construct to perform SAXS is the most direct way to evaluate our crystal structure. However, as shown below, this construct is not the most obvious choice.

We have collected SAXS data of the crystallized construct, the complex of mH2A1.1¹⁸¹⁻³⁶⁶-tagged SPX2^{1-202/Δ47-59} and PHR2²²⁵⁻³⁶². SAXS experiments were performed at three different concentrations, and they harbor identical scattering profiles (Response Fig. 1). This indicates that the association between mH2A1.1¹⁸¹⁻³⁶⁶-tagged SPX2^{1-202/Δ47-59} and PHR2²²⁵⁻³⁶² is concentration independent, and the protein aggregation was less happened. However, the experimental scattering data is significant different from the theoretical scattering profile of the crystal structure (Response Fig. 1). We speculate that the discrepancy may be resulted from two reasons. Firstly, for the calculation of theoretical scattering profile, since we did not calculate the scattering contribution from the invisible residues and the deleted internal-residues in the crystal structure, the discrepancy may be some contributed by these unconsidered residues. Furthermore, the fused mH2A1.1¹⁸¹⁻³⁶⁶ tag may take alternative orientations in solution which is different from the packed conformation in the crystal, leading the discrepancy between the theoretical scattering profile and experimental data. Thus, the crystallized construct is not the most obvious choice to

evaluate the crystal structure in solution by SAXS. Then, for SAXS characterization, we removed the mH2A1.1¹⁸¹⁻³⁶⁶ tag and used the construct of SPX2¹⁻²⁰²/PHR2²²⁵⁻³⁶² to evaluate just the SPX2¹⁻²⁰²/PHR2²²⁵⁻³⁶² part in the mH2A1.1¹⁸¹⁻³⁶⁶-tagged crystal (Supplementary Fig. 6 in previous manuscript).

Regarding to your concerns about our SAXS characterization of the not crystallized protein (Supplementary Fig. 6 in previous manuscript), please find our response in your Major Point 7) that we have added more details and discussion in the revised Methods and revised Supplementary Fig. 6.

Response Fig. 1. SAXS measurement of the crystallized mH2A1.1¹⁸¹⁻³⁶⁶-tagged SPX2^{1-202/Δ47-59}/PHR2²²⁵⁻³⁶² complex. SAXS measurement was performed at 284 μM (black dots), 142 μM (orange dots), 71 μM (green dots), respectively. The three experimental scattering data were scaled by the first point. The theoretical scattering profile of the crystal structure (red line) was calculated using CRY SOL and scaled by the first point of the experimental scattering data.

4) Structure suggests residues from each protomer stabilize IP6. K147A in protomer B alone does not destabilize, nor does R31A alone in protomer A. If these mutations are epistatic and encoded on separate plasmids/alleles, contribution from each protomer should decrease SPX-PHR binding, testing model.

Response: Following your constructive suggestion, we encoded the R31A and / or

K147A mutation(s) on two separate SPX2 molecules to test InsP₆ binding model (Response Fig. 2). We co-expressed SPX2, flag-tagged SPX2 and His-tagged PHR2 in *E.Coli*. (Response Fig. 2a). In the presence of InsP₆, three types of SPX2/InsP₆/PHR2 complex should be formed, including the PHR2 in complex with two kinds of homo SPX2 dimers (SPX2/SPX2 dimer and SPX2-flag/SPX2-flag dimer) and a hetero SPX2 dimer (SPX2-flag/SPX2 dimer). Following by Ni-NTA pull-down and flag pull-down, PHR2-His in complex with SPX2-flag/SPX2-flag homodimer and SPX2-flag/SPX2 heterodimer would be co-eluted (Response Fig. 2a). Indeed, by flag pull-down, the SPX2 was co-eluted with SPX2-flag and PHR2-His (Response Fig. 2b, line 1). This indicate that the SPX2 associates with SPX2-flag to bind PHR2-His and it is consistent with the 2:2 binding model of SPX2 and PHR2.

Then we first introduced R31A/K147A double mutation into SPX2-flag, co-expressed it only with PHR2-His, performed Ni-NTA pull-down and flag pull-down. We found that the double mutation cannot break the SPX2-PHR2 association (Response Fig. 2b, line 6), indicating that the R31 and K147 residues contribute less for InsP₆ sensing than others showed in Figure 2. Thus, none of the R31A and / or K147A mutation(s) on any protomer break SPX2 – PHR2 association (Response Fig. 2b, line 2-5).

Response Fig. 2. Encoding R31A and / or K147 mutation(s) on two distinct SPX molecules to assess the InsP₆-binding dependent SPX2 – PHR2 association.

5) Some test of IP6 inducibility should be done. SEC-SAXS in absence of IP6 should be wildly different if tetramer does not form.

Response: Thank you for your insightful comments. We completely agree with you that, by comparing the SEC, AUC, and SAXS results of SPX2 or SPX2/PHR2 complex in the presence or absence of InsP₆, we will answer whether the dimerization

of SPX2 is induced by InsP₆ binding. However, we cannot get an amount of soluble SPX2 protein in the absence of InsP₆ to perform these experiments. We have stated this issue in Methods: "We have tried various expression systems, such as *E. Coli.*, *insects*, or mammalian cells, to express the SPX2 alone, but no soluble SPX2 protein was obtained". We have to say that this is the limitation of our work, and we can only conclude that the interaction between SPX2 and PHR2 is InsP₆ dependent and they associate with a stoichiometry ratio of 2:2 in vitro, but we do not know whether this conformation is induced the binding of InsP₆. We have discussed this in our revised manuscript in Discussion: "Domain-swapped dimerization could be induced artificially^{33,34}. Since we cannot get soluble SPX2 alone in the absence or presence of InsP₆, we cannot clarify that if the dimerization of SPX2 is induced by InsP₆ binding, and also cannot exclude that the dimerization of SPX2 is induced by the domain swapping. This is the limitation of our studies, and further in plant studies are needed to clarify the SPX2 dimerization in vivo" (paragraph 1 in page 11).

Less Major:

6) Does alphafold2 predict the extended SPX helix 3, when a monomer (without PHR)?

Response: In the Alphafold2 Protein Structure Database, SPX helix 3 is predicted to be not extended (<https://alphafold.ebi.ac.uk/entry/Q6Z784>).

7) AUC in Fig S5 shows ~10% smaller species which should perturb ensemble SAXS radius of gyration, but the fit up in guinier region looks tight. Guinier fit, chi square and residuals should be shown, and AUC discrepancy addressed in text. As is, looks

like something isn't right and there's not enough detail in SAXS provided to know what it is.

Response: Thank you for your very insightful comments. We have carefully revised our manuscript according to your constructive suggestions, as shown below:

Firstly, the protein sample used for AUC is the complex of full-length SPX2¹⁻²⁸⁰ and PHR2²²⁵⁻³⁶². SDS-PAGE gel showed that this sample contained an additional protein band besides the SPX2¹⁻²⁸⁰ and PHR2²²⁵⁻³⁶². It might be resulted from SPX2¹⁻²⁸⁰ proteolysis or some contamination, therefor caused some smaller species in the AUC result. We have included this SDS-PAGE gel in the revised Supplementary Fig 5b. And, we discussed this issue in its figure legend: "The smaller species in the AUC result of **b** may be contributed from some proteolysis or contamination".

Secondly, in revision, we further performed AUC experiment of the protein complex that we used for crystallization, the complex of mH2A1.1¹⁸¹⁻³⁶⁶-tagged SPX2^{1-202/Δ47-59} and PHR2²²⁵⁻³⁶². It showed that the measured molecular weight is also about twice of the sum of mH2A1.1¹⁸¹⁻³⁶⁶-tagged SPX2^{1-202/Δ47-59} and PHR2²²⁵⁻³⁶² (we added a figure as Supplementary Fig. 5a in the revised Supplementary Information). This indicates that the mH2A1.1¹⁸¹⁻³⁶⁶-tagged SPX2^{1-202/Δ47-59} binds to PHR2²²⁵⁻³⁶² also with a stoichiometry ratio of 2:2 in solution, and it is consisted with the crystal structure. This protein sample is homogeneous and only a single species turns out in the AUC result (Supplementary Fig. 5a in the revised Supplementary Information).

Taken together, our AUC results validated the stoichiometry ratio of SPX2/PHR2 complex in solution, and indicated that mH2A1.1¹⁸¹⁻³⁶⁶ fusion tag and the internal-residues deletion have little impact on the SPX2 structure and its ability to bind PHR2.

Thirdly, we are sorry for missing some important details in the SAXS characterization. The rationale behind this SAXS design is that, in order to validate the structure of SPX2¹⁻²⁰²/PHR2²²⁵⁻³⁶² visualized in our crystal, we performed SAXS measurement by using the sample of SPX2¹⁻²⁰²/PHR2²²⁵⁻³⁶² complex that has no mH2A1.1 tag and no

internal-residues deletion. This sample is stable with no proteolysis and contains no contamination (Response Fig. 3), that is different from the AUC sample of full-length SXP¹⁻²⁸⁰/PHR2²²⁵⁻³⁶² complex (Supplementary Fig. 5a in the revised Supplementary Information). Thus, this homogeneous SPX2¹⁻²⁰²/PHR2²²⁵⁻³⁶² complex is suitable for SAXS to assess the structural difference between crystal and solution. The two different samples used for AUC (SPX2¹⁻²⁸⁰/PHR2²²⁵⁻³⁶² with some proteolysis or contamination) and SAXS (homogeneous SPX2¹⁻²⁰²/PHR2²²⁵⁻³⁶²) may be the origin of your impression of "AUC discrepancy" concerns. We are sorry for these missed details in our previous manuscript. We have included these details in the revised Methods and Supplementary Fig. 5 and 6.

Fourthly, following your concerns and suggestions, we included the Guinier fit, residuals and the radius of gyration (Rg) derived from the experimental data in the revised Supplementary Fig. 6. Guinier fitting of low-resolution scattering data ($q < 0.033 \text{ \AA}^{-1}$) determined a Rg of $41.4 \pm 0.5 \text{ \AA}$ with reasonable residuals. To evaluate the crystal structure in solution, the invisible residues and deleted internal-residues in the crystal structure were patched and optimized for the calculation of structure-derived theoretical SAXS profile. For this calculation, the invisible residues (including I35-M66, P191-G202 of SPX2¹⁻²⁰² and S225-T247, L307-G329 of PHR2²²⁵⁻³⁶²) and the deleted internal-residues R47-T59 of SPX2¹⁻²⁰² in the crystal structure of SPX2^{1-202/Δ47-59}/InsP₆/PHR2²²⁵⁻³⁶² complex were patched using PyMOL. Considering the flexibility of these patched residues, this full SPX2¹⁻²⁰²/InsP₆/PHR2²²⁵⁻³⁶² structure was subjected to Xplor-NIH for the optimization of these patched residues. During the optimization, the coordinates for the visible crystal parts were fixed, only these patched residues were given full torsional freedom to minimize total energy. 260 conformations were optimized to account the flexibility of these patched residues, and the theoretical SAXS profiles of this conformation ensemble were calculated and plotted against the experimental data. It shows that there is no significant difference between the calculated profiles and experimental data (Supplementary Fig. 6 in the revised Supplementary Information). The theoretical calculated Rg of this optimized

conformation ensemble is $42.1 \pm 1.0 \text{ \AA}$, that is consistent with the experimental R_g ($41.4 \pm 0.5 \text{ \AA}$). The model we used to fit the SAXS data in our previous manuscript is a conformation that has the lowest energy among the optimized 260 conformations. It fits well with the experimental data of $\text{SPX2}^{1-202}/\text{InsP}_6/\text{PHR2}^{225-362}$ complex in solution. The chi square (χ^2) for the fitting is 3.20 (Supplementary Fig. 6 in the revised Supplementary Information). We have included these information and details in the revised Methods and Supplementary Fig. 6.

In summary, we conclude that the crystal structure of $\text{SPX2}^{1-202}/\text{InsP}_6/\text{PHR2}^{225-362}$ complex is maintained in solution.

Response Fig. 3. The complex of $\text{SPX2}^{1-202}/\text{InsP}_6/\text{PHR2}^{225-362}$ used for SAXS measurement is stable with no proteolysis and contains no contamination.

8) For calculated SAXS, residues missing from the crystal structure were modeled which can introduce bias as its very easy to put the residues wherever they improve SAXS fit. Authors should make it clear the model was not changed iteratively, and without peaking at the fit with observed SAXS.

Response: Thank you for your concerns and suggestions, and we are sorry for missing some important details in our previous manuscript. We have also responded these concerns in your last Major point 7.

For model used to assess SAXS data, the model was first optimized without SAXS fitting and refinement. To calculate the theoretical SAXS profile of $\text{SPX2}^{1-202}/\text{InsP}_6/\text{PHR2}^{225-362}$ complex, the invisible residues (including I35-M66, P191-G202 of SPX2^{1-202} and S225-T247, L307-G329 of $\text{PHR2}^{225-362}$) and the deleted

internal-residues R47-T59 of SPX2¹⁻²⁰² in the SPX2^{1-202/Δ47-59}/InsP₆/PHR2²²⁵⁻³⁶² crystal structure were patched using PyMOL. And, considering the flexibility of these patched residues, this full SPX2¹⁻²⁰²/InsP₆/PHR2²²⁵⁻³⁶² structure was subjected to Xplor-NIH for the optimization of these patched residues. During the optimization, the coordinates for the visible crystal parts were fixed, only these patched residues were given full torsional freedom to minimize total energy. 260 conformations were optimized and one conformation with the lowest energy was used to assess the fitting of the experimental SAXS data in our previous manuscript. In revision, we have included these details in Methods, plotted the theoretical SAXS profiles of all the 260 conformations and added more analyses and discussion in the revised Supplementary Fig. 6.

9) Data in Figure 2C is inaccurately described in line 110, in Figure 2C and in the methods. Authors state IP6 binding is being measured, and provide their explanation, but they do not measure IP6 binding. They only measure SPX-PHR interaction perturbed by mutations at the putative IP6 site. The mutations are the variable, language should be changed to state this. Or they can actually measure IP6 binding using radiolabels and report K_d's, which would be much better.

Response: Thank you for your comments, reminding us that we should be more accurate to describe the "InsP₆ binding" of SPX2.

In revision, for line 110 in our previous manuscript, we used "InsP₆-binding dependent SPX2 – PHR2 association" to describe this relationship, that is "Consequently, the InsP₆-binding dependent SPX2 – PHR2 association was strongly reduced / abolished by the Y25F, Y25A, L28A, K29A, or K143A/K147A substitutions of SPX2 (Fig. 2c), indicating that these mutations have perturbed the sensing of InsP₆ by SPX2." (last paragraph in page 7).

In the revised Figure 2C, we used "InsP₆-binding dependent SPX2 – PHR2 association" to replace "InsP₆ binding". And, we used "we applied a co-expression coupled Ni-NTA pull-down strategy to indirectly assess the recognition of InsP₆ by

monitoring the InsP₆-binding dependent SPX2 – PHR2 association" to replace "we applied a co-expression coupled Ni-NTA pull-down strategy to assess ligand binding and protein interactions" in revised Methods.

Reviewer #2 (Remarks to the Author):

PHR and SPX family proteins are key players in maintaining the homeostasis of Pi. In the presence of inositol pyrophosphates (PP-InsPs), SPX family proteins can interact with PHR proteins, inhibiting their gene activation ability and avoiding the toxicity of high concentration of Pi. In this manuscript, Guan et al. determined the complex structure of rice SPX2/InsP6/PHR2, revealed the detailed interaction between SPX2 and PHR2 and the functional importance of PP-InsPs. In addition, they also determined the crystal structure of PHR2 MYB domain in complex with its target DNA. This work helps to understand the molecular mechanism how SPX2 inhibits PHR2 from binding to the target genes to regulate Pi homeostasis in rice. The major concern is how to interpret that SPX2 formed domain-swapped dimer in the SPX2/InsP6/PHR2 complex.

Response: We are glad to see that you recognize the importance of our work. We truly appreciate your insightful comments and very constructive suggestions that helped us to significantly improve our manuscript. According to your suggestions, we have performed more experiments and analyses. These results and discussions were included in the revised manuscript. Please find our point-by-point response to your concerns listed below.

Concerns:

1) In this work, the authors found that rice SPX2 formed domain-swapped dimer in the ternary complex, which has NOT been previously observed in other SPX domain structures. Although the authors verified the dimerization state of rice SPX2 by Analytical ultracentrifugation (AUC), I still suspect that the dimerization of SPX2 is probably artificial by domain swapping. Please test the dimerization of SPX2 by gel filtration assay to clarify whether the domain swapping is caused by the truncation of PHR2 (225-362). Since the full-length PHR2 is not stable, the authors can check PHR2 (231-426) (Wang et al., PNAS, 2014), full-length SPX2 (1-280), PHR2 (231-426) and SPX2 (1-280) in presence or absence of InsP6 in gel filtration.

If the authors failed to prove that SPX2 could form dimer in gel filtration, their complex structure of rice SPX2/InsP6/PHR2 still makes sense, but they have to clarify that the dimerization state of rice SPX2 in the complex is due to domain swapping.

Response: Thank you for your insightful comments and constructive suggestions, reminding us that we should be more careful to state the "domain-swapped dimer" of SPX2. We have carefully revised our manuscript and included a more rational discussion, as shown below:

Firstly, to exclude the possibility that the mH2A1.1¹⁸¹⁻³⁶⁶ fusion tag and the internal-residues deletion of SPX2 may artificially induce a 2:2 binding model of SPX2 and PHR2, we further performed AUC experiment of the protein complex that we used for crystallization, the complex of mH2A1.1¹⁸¹⁻³⁶⁶-tagged SPX2^{1-202/Δ47-59} and PHR2²²⁵⁻³⁶² (we added a figure as Supplementary Fig. 5a in the revised Supplementary Information). We found that either the protein complex used for crystallization or the complex of full-length SPX2¹⁻²⁸⁰ and PHR2²²⁵⁻³⁶² are both assembled by a 2:2 stoichiometry ratio (Supplementary Fig. 5 in the revised Supplementary Information), consistent with the complex crystal structure. This indicates that the mH2A1.1¹⁸¹⁻³⁶⁶ fusion tag and internal-residues deletion of SPX2 have little impact on the SPX2 structure and its ability to bind PHR2.

Secondly, in revision, we performed more detailed SAXS analysis (Supplementary Fig. 6 in the revised Supplementary Information and Methods) to validate our structure model. SAXS characterization of SPX2¹⁻²⁰²/PHR2²²⁵⁻³⁶² complex (no mH2A1.1¹⁸¹⁻³⁶⁶ tag and no internal-residues deletion) well validated that two PHR2²²⁵⁻³⁶² molecules bind a domain-swapped SPX2¹⁻²⁰² dimer in solution and the conformation of this complex is similar to that visualized in crystal.

Thirdly, we further performed more chemical crosslinking experiments to validate the domain-swapped conformation of SPX2 dimer in the complex of SPX2/InsP₆/PHR2, by encoding the K106C and C182S mutation on two separate SPX2 molecules. We

have included these new results in the revised main text (paragraph 2 in page 6 and paragraph 1 in page 7) and added a figure as Supplementary Fig. 8 in the revised Supplementary Information. These new results collectively conformed that the crosslinking was occurred between two distinct SPX molecules and validated the domain-swapped conformation of SPX2 dimer in the complex of SPX2/InsP₆/PHR2 in solution.

Fourthly, following your constructive suggestions, we constructed the PHR2²³¹⁻⁴⁶² boundary to check if the domain swapping of SPX2 is caused by the truncation of PHR2²²⁵⁻³⁶². Unfortunately, by co-expressing SPX2¹⁻²⁸⁰ and PHR2²³¹⁻⁴²⁶-His, the SPX2¹⁻²⁸⁰ was expressed as inclusion body, and no soluble complex of SPX2¹⁻²⁸⁰/PHR2²³¹⁻⁴²⁶-His was co-eluted from Ni-NTA beads, regardless of the presence of InsP₆ (Response Fig. 1). Although we can get soluble PHR2²³¹⁻⁴²⁶-His, we cannot get soluble SPX2¹⁻²⁸⁰ alone (Response Fig. 1), as well as we stated this issue in our previous manuscript. Thus, we could not perform the gel filtration assay for the SPX2¹⁻²⁸⁰, PHR2²³¹⁻⁴²⁶, and SPX2¹⁻²⁸⁰/PHR2²³¹⁻⁴²⁶ complex in the presence or absence of InsP₆.

In summary, although we believe that we have well validated the domain-swapped conformation of SPX2 dimer in the complex of SPX2/InsP₆/PHR2 in solution, we cannot exclude the possibility that the dimerization of SPX2 is induced by domain swapping. In revision, we have clarify this possibility in Discussion, that is: "Domain-swapped dimerization could be induced artificially^{33,34}. Since we cannot get soluble SPX2 alone in the absence or presence of InsP₆, we cannot clarify that if the dimerization of SPX2 is induced by InsP₆ binding, and also cannot exclude that the dimerization of SPX2 is induced by the domain swapping. This is the limitation of our studies, and further in plant studies are needed to clarify the SPX2 dimerization in vivo." (paragraph 1 in page 11).

Response Fig. 1. Expression and purification of SPX2¹⁻²⁸⁰/PHR2²³¹⁻⁴²⁶-His, PHR2²³¹⁻⁴²⁶-His, SPX2¹⁻²⁸⁰-His in the presence or absence of InsP₆. Supernatant of the lysed cells were loaded on Ni-NTA beads. After washing unbound proteins, the target protein was eluted by imidazole.

2) In panel C of Fig. 2, SPX2 K29A mutant failed to interact with PHR2. However, the side chain of SPX2 K29 is pointed away from InsP₆ in the structure (see Fig. 2 panel A). Similar problem was also observed for SPX2 K143 in Fig. 2 panels A and C. Please check the conformation of K29 and K143 in the structure. In addition to InsP₆, it is also helpful to include the electron density of K29 and other residues critical for InsP₆ recognition in the manuscript.

Response: Thank you for your concerns. We have included the electron densities of these critical residues for InsP₆ binding as Supplementary Fig. 9 in the revised Supplementary Information. As you see, the densities of these residues are poor, that may be due to the low resolution of the structure.

3) In Table S1, please check the B factor for the ternary complex. The overall B-factor is very high (over 147 and 280 Å² for overall structure and other entities). Besides, the B-factor of the side chain atoms is much lower than other atoms.

Response: Thank you for your reminder. The B-factor of side chain atoms is 157, not 15.7. We are sorry for this typing-error in our previous manuscript. We have corrected it in revision.

We have carefully checked the B factors of our structure and uploaded the PDB validation reports in revision. We found that the statistics are most ok. Please find the uploaded PDB validation reports. Regarding to the high B-factor, it may be due to the low resolution of the structure, indicating flexibility of the protein complex.

REVIEWER COMMENTS

Reviewer #1 (Remarks to the Author):

The authors have done an exemplary job of addressing all concerns. My only recommendation is that Response Fig 1 be included in supplemental data to highlight the differences between measured vs. calculated SAXS of the crystalized construct. The discrepancy does not alter any conclusions from the paper but should be included for completeness.

Reviewer #2 (Remarks to the Author):

I know that SPX1 and PHR2 formed a complex with a 1:1 stoichiometry in the presence of InsP6 by SEC-MALS analysis, and a crystal structure of the InsP6-induced 1:1 SPX1-PHR2 complex was also resolved.

So the authors have to prove the InsP6-induced dimerization of SPX2 (not in complex with PHR2) to support their model in Fig.4e.

Even they failed to get soluble SPX2 1-280, they may try more vectors to get soluble SPX2 (SPX1 1-259 is soluble) to make a correct conclusion, which is the most important.

Reviewer #1 (Remarks to the Author):

The authors have done an exemplary job of addressing all concerns. My only recommendation is that Response Fig 1 be included in supplemental data to highlight the differences between measured vs. calculated SAXS of the crystallized construct. The discrepancy does not alter any conclusions from the paper but should be included for completeness.

Response: Thank you for your comment. We have now included the SAXS data and discussion of the crystallized construct in the revised manuscript (paragraph 1 on page 5 and Supplementary Fig. 6a in the revised Supplementary Information).

Reviewer #2 (Remarks to the Author):

I know that SPX1 and PHR2 formed a complex with a 1:1 stoichiometry in the presence of InsP₆ by SEC-MALS analysis, and a crystal structure of the InsP₆-induced 1:1 SPX1-PHR2 complex was also resolved.

So the authors have to prove the InsP₆-induced dimerization of SPX2 (not in complex with PHR2) to support their model in Fig.4e.

Even they failed to get soluble SPX2 1-280, they may try more vectors to get soluble SPX2 (SPX1 1-259 is soluble) to make a correct conclusion, which is the most important.

Response: Thank you for pointing this out. We completely agree with you that it is necessary to further characterize the InsP₆-induced SPX2 dimerization to support the model shown in Fig.4e. Actually, to clarify if InsP₆ can induce SPX2 dimerization, during this project we have tried many methods to try to get an amount of soluble SPX2 alone, by using lots of SPX2 boundaries, expression vectors and systems. Unfortunately, we failed.

While our last revised manuscript was under review, another work reported the crystal structure of rice SPX1¹⁻¹⁹⁸ in complex with PHR2²⁴⁸⁻³⁸⁰ and InsP₆ (Nat Commun 12(1):7040 doi: 10.1038/s41467-021-27391-5). In that work, the authors got soluble SPX1 alone. And thank you for your suggestion, we constructed the borders of the reported soluble SPX1¹⁻²⁵⁹ and SPX1¹⁻¹⁹⁸ onto SPX2 (corresponding to SPX2¹⁻²⁵⁷ and SPX2¹⁻¹⁹⁴, respectively). However, although the SPX1 fragment is soluble, the SPX2 fragment is still insoluble (Response Fig. 1). We also tested more boundaries and tried different vectors, but none of them worked (Response Fig. 1).

While our last revised manuscript was under review, we continued to try to obtain soluble SPX2. At last, we got a soluble SPX2^{1-267/Δ47-64} fragment that including residues 1-267 with a deletion of residues 47-64 (Supplementary Fig. 13 in the revised Supplementary Information and the revised Methods). SEC and AUC analysis revealed

that InsP₆ binding does not induce SPX2 dimerization in solution. Thus, we modified the model shown in Fig. 4e, by removing the part of InsP₆-induced SPX2 dimer.

Additionally, we found that no-InsP₆-bound SPX2^{1-267/Δ47-64} harbours a broad SEC profile and inhomogeneous AUC species, indicating its inhomogeneous conformations/oligomeric states. InsP₆ binding appears to stabilize a particular conformer. This is consistent with the reported results in which InsP₆ binding stabilizes the α1 of rice SPX1. Furthermore, we sought also to analyze InsP₆-induced conformational changes of other plant stand-alone SPX proteins. By expressing all the full-length stand-alone SPX proteins from rice and *Arabidopsis* using *E.Coli.*, we obtained soluble AtSPX2, AtSPX4, OsSPX1 and OsSPX4 (Response Fig. 2). SEC assay revealed different behaviours of these SPX proteins upon InsP₆ binding, indicating that the plant SPX receptors may undergo different conformational changes upon PP-InsPs sensing. We have included these data and discussion in Discussion in our revised manuscript, and we added three figures as Supplementary Figs. 12-14 in the revised Supplementary Information.

Response Fig. 1. Expression test of OsSPX2 using different boundaries and vectors. Supernatant of the lysed cells were loaded on Ni-NTA beads. After washing unbound proteins, the target protein was eluted by imidazole. While the OsSPX1¹⁻²⁵⁹ can be soluble expressed (highlighted in red), the equal bondor of OsSPX2¹⁻²⁵⁷ was expressed in pellet (highlighted in magenta). Soluble and insoluble target proteins were highlighted with red triangle and magenta triangle, respectively.

Response Fig. 2. Expression of AtSPX1, AtSPX2, AtSPX3, AtSPX4, OsSPX1, OsSPX3, OsSPX4, OsSPX5 and OsSPX6 using pET15 D or pET21 B vector, respectively. Supernatant of the lysed cells were loaded on Ni-NTA beads. After washing unbound proteins, the target protein was eluted by imidazole. Soluble target proteins were highlighted with a red triangle.

REVIEWERS' COMMENTS

Reviewer #2 (Remarks to the Author):

The authors have addressed all my concerns.